# Faking Interpolation Until You Make It

**Alasdair Paren**                                        *alasdair.paren@gmail.com*
*Department of Engineering Science*
*University of Oxford*
*Oxford, UK*

**Rudra P. K. Poudel**                              *rudra.poudel@crl.toshiba.co.uk*
*Cambridge Research Laboratory,*
*Toshiba Europe Ltd,*
*Cambridge, UK.*

**M. Pawan Kumar**                                    *pawan@robots.ox.ac.uk*
*Department of Engineering Science*
*University of Oxford*
*Oxford, UK.*

**Reviewed on OpenReview:** *https://openreview.net/forum?id=OslAMMF4ZP*

## Abstract

Deep over-parameterized neural networks exhibit the interpolation property on many data sets. Specifically, these models can achieve approximately zero loss on all training samples simultaneously. This property has been exploited to develop optimisation algorithms for this setting. These algorithms use the fact that the optimal loss value is known to employ a variation of a Polyak step size calculated on each stochastic batch of data. We introduce a novel extension of this idea to tasks where the interpolation property does not hold. As we no longer have access to the optimal loss values *a priori*, we instead estimate them for each sample online. To realise this, we introduce a simple but highly effective heuristic for approximating the optimal value based on previous loss evaluations. We provide rigorous experimentation on a range of problems. From our empirical analysis we demonstrate the effectiveness of our approach, which outperforms other single hyperparameter optimisation methods.

## 1 Introduction

Deep over-parameterized neural networks exhibit the interpolation property on many data sets (**?**Berrada et al., 2020). That is, these models are able to achieve close to zero loss on all samples simultaneously. Initially, the interpolation property has been exploited to prove new convergence rates (Vaswani et al., 2019a;b; Ma et al., 2018; Liu & Belkin, 2019) and to develop novel optimisation algorithms for this setting. Examples include Adaptive learning rates for Interpolation with Gradients (ALI-G; Berrada et al. (2020)), and Stochastic Polyak Step (SPS; Loizou et al. (2021)). Both ALI-G and SPS use the interpolation property to ensure that the optimal loss value for each data point will be zero. With this knowledge it is possible to employ a stochastic variation of the Polyak step size (Polyak, 1969). This automatically scales a maximal step size hyperparameter down to an appropriate value for each update, which removes the need for a painstakingly hand-designed learning rate schedule (Berrada et al., 2020; Loizou et al., 2021). ALI-G and SPS have been shown to produce highly competitive results, matching the generalisation performance of SGD with a manually tuned learning rate in many settings, and outperforming adaptive gradient methods by a large margin. While these techniques work well, the interpolation property does not hold on many interesting large-scale learning tasks, or in situations where the model size is limited.

In this work, we propose a novel optimisation method for non-interpolating problems inspired by algorithms designed for interpolation. Our approach is based on the observation that any non-interpolating problem can be made to satisfy the interpolation property once a point that minimises the training objective is known. One simply modifies each loss to be the point-wise minimum of the loss function and its value at the optimal point. Moreover, one only requires the knowledge of this optimal loss value for every example and not the location in parameter space of the minimiser. Hence, if one is able to approximate the loss values at an optimal point with reasonable accuracy, one should be able to replicate the desirable characteristics of algorithms such as ALI-G and SPS. Specifically, we will be able to obtain an algorithm with a single fixed hyperparameter that is easy to tune, and has a strong generalisation performance. We present an optimisation method that approximates the optimal function values online using a heuristic in combination with a Polyak step size. We name our algorithm Adaptive ALI-G (ALIG+), as it makes use of ALI-G iteratively to update the parameters.

We conduct a thorough empirical evaluation of ALI-G+ on a variety of tasks against strong baselines. We provide results for matrix factorisation, binary classification using RBF kernels, image classification on the SVHN, CIFAR, Tiny ImageNet and ImageNet datasets, and review classification and next character prediction. These tasks are designed to provide a mix of non interpolating and interpolating problems, in all cases ALI-G+ outperforms all other single hyperparameter method often by a significant margin. These results demonstrate that estimating the optimal loss value online is an effective alternative approach for selecting the step size.

## 2 Related Works

We discuss existing optimisation methods for supervised learning tasks that do not satisfy the interpolation property. The approaches can be broadly classified into three categories: SGD with a manually tuned learning rate schedule, line search methods, and adaptive gradient methods.

**SGD with a Learning Rate Schedule.** SGD (Robbins & Monro, 1951) has been used to produce state of the art performance for supervised learning tasks. However, the downside of SGD is that it requires the manual design and refinement of a learning rate schedule for best performance. Many forms of schedule have been proposed in the literature, including piecewise constant (Huang et al., 2017), geometrically decreasing (Szegedy et al., 2015) and warm starts with cosine annealing (Loshchilov & Hutter, 2017). Consequently, practitioners who wish to use SGD in a novel setting need to select which type of schedule to use for their learning task. To that end, they first need to choose the parameterization of the schedule and then tune the corresponding hyperparameters. For example, a piecewise linear scheme requires an initial learning rate value, a decay factor and a list or metric to determine at which points in training to decay the learning rate. This results in a large search space which increases exponentially in combination with other problem dependent quantities such as regularisation amount or batch size. As SGD can be sensitive to these hyperparameters, and their optimal values often are highly interdependent, the resulting cross-validation scheme necessary for best results can be prohibitively expensive.

**Line Search Methods.** Line search methods, such as those developed by Vaswani et al. (2019b); Mutschler & Zell (2020); Hao et al. (2021) offer an appealing alternative to SGD as they remove the need to find a learning rate schedule and instead run extra forward passes to select a step size. While not specifically designed for settings where interpolation does not hold, Vaswani et al. (2019b) present algorithms based around the Armijo and Goldstein line-search methods, classically used for deterministic gradient descent. They also introduce heuristics with the aim of minimising the number of extra forward passes required, which they claim reduces the average number required to one per batch. Mutschler & Zell (2020) and Hao et al. (2021) instead assume the loss function is approximately parabolic in the negative gradient direction, and thus use extra forward passes to construct a parabolic model of the loss that can then be minimised in closed form. Mutschler & Zell (2020) additionally provide empirical justification for the parabolic approximation. While line-search methods present strong performance they invariably introduce extra hyperparameters governing how points are selected in the line search or whether a target point is accepted. While these hyperparamters are held fixed over training and do not require a schedule they must be tuned per experiment for best

results. Furthermore, line search methods require approximately twice the computation per batch of typical first order methods resulting in a far longer training time.

**Adaptive Gradient Methods** Adaptive gradient methods such as Adagrad (Duchi et al., 2011), Adam (Kingma & Ba, 2015) or more recently Adabound (Luo et al., 2019) use heuristics based on previous gradient evaluations to scale a learning rate for each parameter independently. These algorithms are easy to use as they require a single fixed learning rate hyperparameter that tends to provide decent results over a wide range of values (Sivaprasad et al., 2020). However, once tuned, non-adaptive optimisation algorithms such as ALI-G and SGD provide superior generalisation performance over adaptive gradient methods on a range of supervised learning benchmarks (Berrada et al., 2020; Wilson et al., 2017).

## 3 Preliminaries

**Loss Function.** As is standard for supervised learning, we consider tasks where the model is parameterized by $\boldsymbol{w} \in \mathbb{R}^d$. We assume the objective function can be expressed as an expectation over $z \in \mathbb{Z}$, where $z$ is a random variable indexing the samples of the training set $\mathbb{Z}$:

$$f(\boldsymbol{w}) \triangleq \mathbb{E}_{z \in \mathbb{Z}}[\ell_z(\boldsymbol{w})]. \tag{1}$$

Here $\ell_z$ is the loss function associated with the sample $z$. We assume that each $\ell_z$ admits a known lower bound $B$. For the large majority of loss functions used in machine learning, such as cross-entropy or hinge losses, the lower bound is $B = 0$. However, we do not assume this lower bound is reached during training. In other words, the interpolation property does not hold.

**Learning Task.** We consider the task of finding a feasible vector $\boldsymbol{w}_\star \in \Omega$ that minimises $f$:

$$\boldsymbol{w}_\star \in \operatorname*{arg\,min}_{\boldsymbol{w} \in \Omega} f(\boldsymbol{w}) + \frac{\lambda}{2}||\boldsymbol{w}||^2, \tag{$\mathcal{P}$}$$

where $\lambda$ controls the regularisation amount. We used weight decay for convenience as it allows for simple comparison with other algorithms. However, ALI-G+ can easily be used with other forms of regularisation. For unconstrained problems, like those considered in this paper, we set to $\Omega = \mathbb{R}^d$.

**The ALI-G Algorithm.** ALI-G+ is inspired by the ALI-G algorithm (Berrada et al., 2020), hence we formally introduce this method here. ALI-G was designed for the optimisation of interpolating problems, that is, problems where $f(\boldsymbol{w}_\star) = B = 0$. This condition also trivially implies that $\ell_z(\boldsymbol{w}_\star) = 0, \forall z \in \mathbb{Z}$. In order for the interpolation assumption to hold for ($\mathcal{P}$) when regularisation is used, ALI-G does not apply it in the conventional way, and hence $\lambda = 0$. Instead, ALI-G makes use of a constraint based regularisation, where the feasible set is defined as $\Omega = \{\boldsymbol{w} \in \mathbb{R}^d : \|\boldsymbol{w}\|_2^2 \leq r\}$ and $r$ controls the regularisation level. In order to ensure that only feasible solutions are found the iterate is projected back onto the set $\Omega$ after each parameter update. At time step $t$ a sample $z_t$, or in practice a mini-batch, is sampled from $\mathbb{Z}$ and the loss and gradient is evaluated at the current interate $\boldsymbol{w}_t$. ALI-G then selects $\boldsymbol{w}_{t+1}$ as the solution to following the proximal problem:

$$\operatorname*{arg\,min}_{\boldsymbol{w} \in \Omega} \left\{ \frac{1}{2\eta} \|\boldsymbol{w} - \boldsymbol{w}_t\|^2 + \max\{0, \ell_{z_t}(\boldsymbol{w}_t) + \boldsymbol{g}_t^\top (\boldsymbol{w} - \boldsymbol{w}_t)\} \right\}, \tag{2}$$

where $\boldsymbol{g}_t \triangleq \nabla_{\boldsymbol{w}} \ell_{z_t}(\boldsymbol{w}_t)$ and $\eta$ is the step size hyperparameter. This proximal problem is identical to that solved in closed form by the SGD update, with a minor modification. The problem (2) additionally includes a point-wise maximum between the linear approximation of the loss and the known lower bound ($\ell_z(\boldsymbol{w}_\star) = 0$). The dual of (2) is a maximisation over a concave function in one dimension constrained to the interval $[0, 1]$. Hence, one can obtain the optimal point by projecting the unconstrained solution onto the feasible region. After some simplification this results in the following closed form solution:

$$\boldsymbol{w}_{t+1} = \boldsymbol{w}_t - \gamma_t \nabla_{\boldsymbol{w}} \ell_{z_t}(\boldsymbol{w}_t), \tag{3}$$

$$\gamma_t \triangleq \max \left\{ \min \left\{ \eta, \frac{\ell_{z_t}(\boldsymbol{w}_t) - \ell_z(\boldsymbol{w}_\star)}{\|\nabla_{\boldsymbol{w}} \ell_{z_t}\|^2} \right\}, 0 \right\}. \tag{4}$$

This update can be viewed as a stochastic analog of the Polyak step size (Polyak, 1969), with the addition of a maximal value $\eta$. From the interpolation assumption, we have $\ell_z(\boldsymbol{w}_\star) = 0, \forall z \in \mathbb{Z}$ and hence the numerator of the fraction in (4) can be simplified to $\ell_{z_t}(\boldsymbol{w}_t)$. Additionally the maximum with zero is redundant as both numerator and denominator of the fraction will always be positive due to the non-negative nature of the loss function. We show both redundant pieces of notation here as it allows us to clearly specify our modified version in the next section. Additionally in practice ALI-G is often used with a Nesterov momentum term for best results. The ALI-G update is computationally cheap with the project on to $\Omega$ and the evaluation of the norm of the gradients being the only extra computation required over SGD. Importantly, ALI-G removes the need for a learning rate schedule and performs comparably on many benchmarks (Berrada et al., 2020).

## 4 Training in Non-Interpolating Settings

While the interpolation setting has received a lot of attention, many interesting problems do not satisfy this assumption. This could be for any of the following reasons: i) the model size could be limited due to hardware or power constraints, such as for embedded devices; ii) the data set is very large, for example, the vast majority of models trained on the ImageNet data set (Deng et al., 2009) do not achieve zero training loss; iii) complexity of the loss function, such as in adversarial training; iv) label noise can make interpolation impossible by creating one to two mappings between inputs and labels. Thus, we think this setting is deserving of bespoke optimisation algorithms that are easy to use and produce strong generalisation performance.

**Motivation.** Our algorithm is motivated by trying to approximate $\ell_z(\boldsymbol{w}_\star)$ online, and as a result recover interpolation. Thus, we introduce a scalar $\tilde{\ell}_z^k$ to store our estimate for each example in the training set. We refer to these scalars as approximate optimal values (AOVs) and the superscript $k$ indicates how many times the approximation has been updated. Our algorithm alternates between two "steps" i) using the current approximation of the optimal loss $\tilde{\ell}_z^k$ to inform the step size, (see Algorithm 1); and ii) improving the approximations based on the best previous iterates, (see Algorithm 2). We describe these "steps" in detail in the following two sections.

**Updating the Parameters.** ALI-G+ uses the same stochastic version of the Polyak step size as ALI-G (Berrada et al., 2020), however, we replace the optimal loss value $\ell_z(\boldsymbol{w}_\star) = 0$ with its current approximation $\tilde{\ell}_z^k$. Hence, at time $t$ the ALI-G+ algorithm uses the following weight update:

$$\boldsymbol{w}_{t+1} = \boldsymbol{w}_t - \gamma_t \boldsymbol{g}_t', \tag{5}$$

$$\gamma_t \triangleq \max\left\{\min\left\{\eta, \frac{\ell_{z_t}(\boldsymbol{w}_t) - \tilde{\ell}_z^k}{\|\nabla_{\boldsymbol{w}} \ell_{z_t}\|^2}\right\}, 0\right\}. \tag{6}$$

We define $\boldsymbol{g}_t' \triangleq (\nabla_{\boldsymbol{w}} \ell_{z_t}(\boldsymbol{w}_t) + \lambda \boldsymbol{w}_t)$, where $\eta$ and $\lambda$ are the hyperparameters controlling the maximum step size and weight decay amount, respectively. The loss, AOV and gradient values are average over the batch, see Appendix A for explicit step size formulae for mini-batches. As we do not require the interpolation assumption to hold, we do not need to use the constraint based regularisation of ALI-G, and can simply make use of weight decay which allows for easy comparison with other algorithms. It is worth noting here that the max with 0 is no longer redundant as there is no guarantee that $(\ell_{z_t}(\boldsymbol{w}_t) - \tilde{\ell}_z^k)$ will be positive. Without this positivity constraint a negative step size could be used resulting in a gradient ascent step. Moreover if $\ell_{z_t}(\boldsymbol{w}_t)$ is already lower than its AOV $\tilde{\ell}_z^k$ then we have already completed our goal for this sample in this "step". In the next "step" we can then update this AOV to a lower value, which we describe in the next section. The full procedure for updating the parameters given the AOVs is outlined in Algorithm 1. In Appendix B we detail some unsuccessful parameter update schemes.

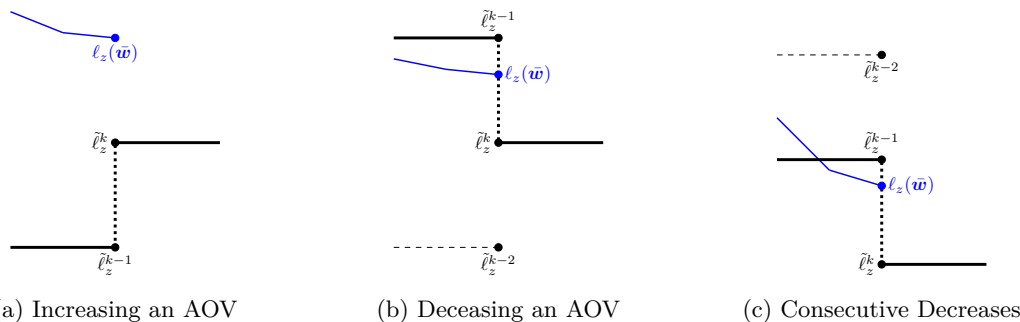

(a) Increasing an AOV        (b) Deceasing an AOV        (c) Consecutive Decreases

Figure 1: The possible AOV updates of a single sample. Thick black line represents AOV values. Blue line depicts $\ell_z(\bar{\boldsymbol{w}})$, Thin black line represents previous AOV value. Panel (a) Shows the update if and AOV has not been reached. Panel (b) conversely shows the process for a AOV that has been reached for the first time, the AOV is lowered half way to the previous value. (c) For samples that have reach their AOV for consecutive sections the same absolute decrease in value is applied.

---

**Algorithm 1** ALI-G with AOVs

---

1: **Input**: time horizon $T$, initial point $\boldsymbol{w}_0$, maximum step size $\eta$, AOVs $\tilde{\boldsymbol{\ell}}^k$ and $\lambda$.
2: **for** $t = 0, ..., T-1$ **do**
3:      Sample $z_t \in \mathbb{Z}$, $\ell_{z_t}(\boldsymbol{w}_t)$, $\nabla_{\boldsymbol{w}}\ell_{z_t}(\boldsymbol{w}_t)$
4:      Set $\gamma_t = \max\left\{\min\left\{\eta, \frac{\ell_{z_t}(\boldsymbol{w}_t)-\tilde{\ell}_z^k}{\|\nabla_{\boldsymbol{w}}\ell_{z_t}\|^2}\right\}, 0\right\}$
5:      $\boldsymbol{w}_{t+1} = \boldsymbol{w}_t - \gamma_t(\nabla_{\boldsymbol{w}}\ell_{z_t}(\boldsymbol{w}_t) + \lambda\boldsymbol{w}_t)$
6: Return $\bar{\boldsymbol{w}} \approx \arg\min_{t\in\{1,...,T\}}\{f(\boldsymbol{w}_t)\}$

---

**Updating the AOVs.** To replicate the performance of algorithms for interpolation we want the approximation $\tilde{\ell}_z^k$ to tend towards $\ell_z(\boldsymbol{w}_\star)$ throughout training. Due to the stochastic and non-convex nature of training neural networks it is impossible to guarantee this behaviour. However, we present a simple scheme for updating the AOVs that demonstrates strong empirical performance as shown in Section 5. This scheme is inspired by both the curriculum learning literature and the work of Hazan & Kakade (2022), which present a theory for a similar scheme for the convex and deterministic settings. The AOV update scheme is designed to be both reactive and optimistic. By reactive we mean that if a specific sample returns a constant loss its AOV will tend toward this value. By optimistic we mean that the AOVs are updated to a lower value than the current best loss value for this sample, in the hope that a further decrease in loss is possible. In practice the loss values of specific samples can fluctuate throughout training especially when data augmentation is used. However, the step size is calculated on a batch of data, and hence is relatively robust to this noise.

The AOV update scheme is as follows: we store the vectors $\tilde{\boldsymbol{\ell}}^k, \tilde{\boldsymbol{\ell}}^{k-1}, \boldsymbol{\ell}(\bar{\boldsymbol{w}})$ containing $(\tilde{\ell}_z^k, \tilde{\ell}_z^{k-1}, \ell_z(\bar{\boldsymbol{w}}))\forall z \in \mathbb{Z}$, where $\bar{\boldsymbol{w}} = \arg\min_{t\in\{0,...,T\}}\{f(\boldsymbol{w}_t)\}$ in memory. The AOVs $\tilde{\ell}_z^k, \tilde{\ell}_z^{k-1}$ are initialised to our known lower bound on the loss $B$. The training duration is split into $K$ equal sections each with length $T$. During each of these sections we keep the AOVs fixed and try to get a good estimate of $\ell_z(\bar{\boldsymbol{w}})$ for each example. After each of the $K$ sections we update all AOVs simultaneously. Each AOV is updated depending on whether it has been "reached", that is, if $(\ell_z(\bar{\boldsymbol{w}}) \leq \tilde{\ell}_z^k)$ is true. In both cases we are optimistic that the loss can be decreased further from its current value. Hence, if an AOV hasn't been reached it is updated by simply averaging $\ell_z(\bar{\boldsymbol{w}})$ and $\tilde{\ell}_z^k$, see Figure 1 (a). This increases this AOV to halfway between the loss at the best point visited and its current value. However, if $(\ell_z(\bar{\boldsymbol{w}}) \leq \tilde{\ell}_z^k)$ we instead try decreasing $\tilde{\ell}_z^k$ halfway to the last value that was reached $\tilde{\ell}_z^{k-1}$, Figure 1 (b). If the $z^{th}$ AOV is reached again in successive sections we reduce $\tilde{\ell}_z^k$ each time by the same magnitude, see Figure 1 (c). Thus, even if an AOV is incorrectly updated to a value higher than $\ell_z(\boldsymbol{w}_\star)$ it can easily be corrected by consecutive reductions. Lastly we ensure AOVs are never decreased below the lower bound B. Hence, for non-negative losses AOVs are always positive.

This scheme can be thought of as defining a curriculum for the non-interpolating setting. Rather than first focusing on easy examples, at the beginning all examples are treated equally. After time, examples that are identified as hard are given less importance, and the optimiser does not try to optimise these example further. As we know we cannot achieve zero loss on all sample simultaneously it makes sense to not focus over the hardest examples. In Appendix B we detail some unsuccessful AOV update schemes.

---

**Algorithm 2** ALI-G+ Algorithm

---
1: **Input**: time horizon $T_{max}$, $K = 5$, $\bar{w}_0$ and $\tilde{\ell}_z^1, \tilde{\ell}_z^0 = B, \forall z \in \mathbb{Z}$ and $\lambda$.
2: **for** epoch $k = 1, ..., K$ **do**
3:      Run Algorithm 1 with $\bar{w}^{k-1}$, $\frac{T_{max}}{K}$, $\tilde{\boldsymbol{\ell}}^k$, $\eta$ and $\lambda$ to obtain $\bar{w}^k$.
4:      **for** $z \in \mathbb{Z}$ **do**
5:          **if** $\ell_z(\bar{w}) \leq \tilde{\ell}_z^k$ **then**
6:              $\tilde{\ell}_z^{k+1} \leftarrow \max\{\frac{\tilde{\ell}_z^k + \tilde{\ell}_z^{k-1}}{2}, \mathrm{B}\}$
7:              $\tilde{\ell}_z^k \leftarrow \tilde{\ell}_z^{k+1} - \tilde{\ell}_z^{k-1}$,
8:          **else**
9:              $\tilde{\ell}_z^{k+1} \leftarrow \frac{\tilde{\ell}_z^k + \ell_z(\bar{w}^k)}{2}$
10:             $\tilde{\ell}_z^k \leftarrow \tilde{\ell}_z^{k-1}$
11: **Return** $\bar{w}_K$

---

**Implementation Details.** For the above scheme to work well, it is important that the AOVs are not updated too frequently, as this can lead to them trending towards $\ell_z(\bar{w})$ too fast. However, it is also important that the AOVs are updated a sufficient number of times so they can approximate $\ell_z(w_\star)$, if $\ell_z(w_\star)$ is large. We find $K = 5$ provides a good balance between these considerations and fix $K$ to this value. However, ALI-G+ is relatively robust to the choice of $K$ and produces good results for $k \in [5, 10]$. In the Appendix D we provide additional results for $K \in \{3, 10, 20\}$. Furthermore, to save computation we i) avoid calculating $f(w_t)$ exactly and instead approximate this online during each epoch and ii) we use $w_T^{k-1}$ in the place of $\bar{w}_{k-1}$ in line 3 of Algorithm 2. This results in ALI-G+ having a similar run time to SGD, where the only extra computation is the updating of the vectors $\tilde{\boldsymbol{\ell}}^k, \tilde{\boldsymbol{\ell}}^{k-1}, \boldsymbol{\ell}(\bar{w})$ and evaluating the norm of the gradients. As is common practice we report the results of the model with the best validation performance found during training.

**Data Augmentation.** Data augmentation can be thought of in two ways. First, it increases the size of the data set by adding new examples that are simply transformed versions of others. Second, it makes online alterations to the original number of examples. As ALI-G+ is designed for the optimisation of non-interpolating problems, which often have large data sets, we choose to view data augmentation in the second way and save only a single AOV for all possible augmentations. When viewing data augmentation in the first way, training regimes where the number of epochs is less than the number of possible transformations would only visit each example less than once on average. Hence, approximating the optimal value would be challenging. Moreover, for many common data augmentation transforms, such as random crops of images, we would expect the optimal loss value to be highly correlated between the same example under different versions of the transformation. To support this claim we calculate the loss value of all possible crops for a subset of 5000 images chosen from a selection of common data sets. We find empirically, at the start of training that the variance between loss values is on average 20 times lower for the different crops of the same image compared to randomly chosen images. Over training we observe this ratio drops to 5 times lower.

**Justification.** ALI-G+ shares many similarities with the ALI-G Algorithm, moreover before the first AOV update these methods use identical descent directions and step sizes. Thus it is likely one can construct a worse case bounds for ALI-G+ similar to those proposed by Berrada et al. (2020) for settings where $\ell_z(w_\star)$ is bounded above. However the main goal of this paper is to demonstrate the strong empirical performance of ALI-G+ on a broad range of deep learning tasks, where ALI-G+ performs best out of all other single hyperparameter methods considered.

# 5   Experiments

In this section we test the hypothesis that a Polyak like step size in combination with AOVs can produce high accuracy models in the non-interpolating setting. We investigate this through rigorous experiments comparing ALI-G+ against a wide range of single hyperparameter optimisation algorithms on a variety of problems.[1] The problems include both non-interpolating and interpolating settings. We start with relatively simple problems such as matrix factorisation and binary classification using RBF kernels. We then consider the training of deep neural networks on popular image classification benchmarks. Here we use small models for two reasons: i) to induce non-interpolation and ii) to be able to investigate a wider range of data sets, hyperparameters and baselines. Ideally we would have included results for large models trained on even larger data sets, to ensure none interpolation. However, this was not within the limits of our computational resources. We have tried to perform as thorough a set of experiments as possible and our experimental setup is on par with, in fact often exceeds, prior work in this area of machine learning in terms of data sets and baselines considered (Vaswani et al., 2019b; Loizou et al., 2021).

We show that our approach scales to large problems by providing results on the ImageNet data set. Furthermore, we do not just show results for computer vision data sets but also for two NLP tasks, highlighting the flexibility of our approach. We include a wider section of tasks and optimiaiton baselines than many easiler works Vaswani et al. (2019b) All experiments are conducted in PyTorch (Paszke et al., 2017) and are performed on a single GPU except for the ImageNet experiments that use two.

## 5.1   Simple Optimisation Benchmarks

**Setting.**   We first demonstrate the performance of ALI-G+ on matrix factorisation and RBF Binary Classification using the tasks detailed in **?**. The matrix factorisation task can be expressed as:

$$\min_{\boldsymbol{W}_1, \boldsymbol{W}_2} \mathbb{E}_{\mathbf{x} \in \mathbb{X}}[||\boldsymbol{W}_1 \boldsymbol{W}_2 \mathbf{x} - \boldsymbol{A}\mathbf{x}||^2], \tag{7}$$

where $\mathbb{X}$ is a data set of 1000 examples drawn from $\mathcal{N}(\mathbf{x}; 0, \boldsymbol{I})$, $\boldsymbol{A} \in \mathbb{R}^{10 \times 6}$ is randomly generated to have condition number $10^{10}$, $\boldsymbol{W}_1 \in \mathbb{R}^{10 \times k}$, $\boldsymbol{W}_1 \in \mathbb{R}^{k \times 6}$, $\boldsymbol{A}$. The rank of the factorisation $k$ is selected to be one of four different values resulting in two problems where interpolation holds and two where it does not. The binary classification with radial basis functions tasks use the mushrooms and ijcnn dataset from the LIBSVM library of SVM problems (Chang & Lin, 2011). The mushrooms dataset satisfies the interpolation assumption, whereas ijcnn does not.

**Method.**   We compare ALI-G+ against Parabolic Approximation Line Search (PAL) (Mutschler & Zell, 2020) and a selection of the optimisation methods used in **?**. We additionally reuse their code for the baselines. These optimisation algorithms contain a collection of strong line search and adaptive gradient methods, all of which do not require a learning rate schedule. Additionally, the majority have a single step size hyperparameter which makes for fair comparison with ALI-G+ .

**Results.**   The results of these experiments are shown in Figures 2 and 3. On the non-interpolating tasks (rank 1 and rank 4 matrix factorisation and binary classification on the ijcnn data set), ALI-G+ performs comparably to the best algorithms, specifically, PAL (Mutschler & Zell, 2020) and SLS (**?**). On the interpolating tasks ALI-G+ fails to minimise the training loss to machine precision like PAL and SLS. However, it attains the same validation performance. which is impressive as ALI-G+ is not designed for the interpolation setting. For these tasks using ALI-G would be a better option.

## 5.2   Image Classification Experiments

**Setting.**   We run experiments on a broad range of image classification benchmarks. Specifically we use the SVHN (Netzer et al., 2011), CIFAR10, CIFAR100 (Krizhevsky, 2009) and Tiny ImageNet data sets. The SVHN and CIFAR data sets are comprised of 32x32 pixel RGB images. For the SVHN data set we use the

---

[1]code available at https://github.com/Alasdair-P/alig__plus

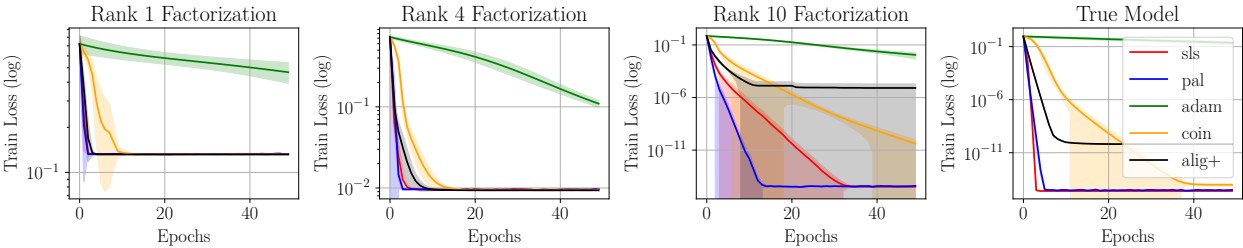

Figure 2: Training performance on the matrix factorisation problem of Vaswani et al. (2019b). In the settings where interpolation does not hold, namely the Rank 1 and Rank 4 problems, ALI-G+ quickly achieves the loss floor. For the Rank 10 and True model problems ALI-G+ does not minimise the loss to machine precision such as SLS (Vaswani et al., 2019b) and PAL (Mutschler & Zell, 2020). Due to the short number of epochs The AOV are updated away from their optimal value of zero before close to zero loss is achieved. The update scheme also fails to decreased the AOVs back to this value. However, ALI-G+ still provides rapid optimisation to at worst $10^{-4}$.

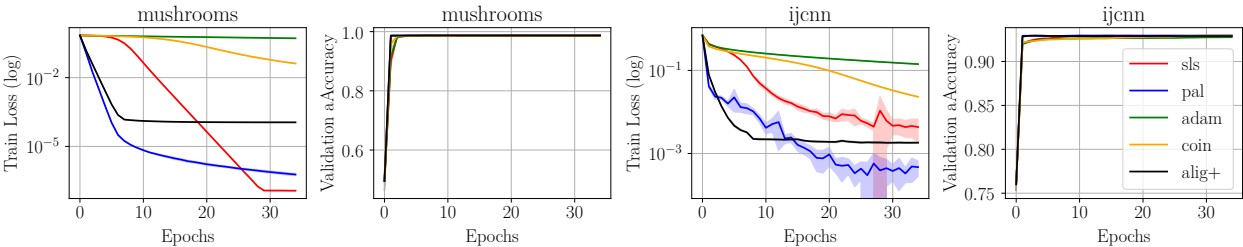

Figure 3: Training and validation performance on the mushrooms and ijcnn data sets (Chang & Lin, 2011). On the mushroom data set, where interpolation holds, ALI-G+ fails to achieve the same training loss as the line search methods. However, in both non-interpolating and interpolating settings ALI-G+ obtains equally good validation performance as the best baseline.

split proposed in Berrada et al. (2020) resulting in 598k training, 6k validation and 26k test samples. SVHN and CIFAR10 both have 10 classes and CIFAR100 has 100. The Tiny ImageNet data set is more challenging and contains 100K training examples of 64x64 pixels split over 200 classes. For the Tiny ImageNet data set the ground truth labels of the test set are not freely available so we report validation scores instead. All images are normalised per channel and when data augmentation is used we apply standard random flips and crops. For the majority of data sets we present results with and without data augmentation. The exceptions being SVHN, which is not designed for data augmentation. For all data sets we make use of the cross entropy loss to train a small 8 layer ResNet (He et al., 2016) containing 90K parameters with 16 channels in the first layer. These tasks were chosen to give examples of i) interpolation (SVHN); ii) near interpolation (CIFAR10) and iii) non-interpolation resulting from limited model size (CIFAR100 and Tiny ImageNet).

**Method.** We compare ALI-G+ against PAL (Mutschler & Zell, 2020), ALI-G (Berrada et al., 2020), AdamP (Heo et al., 2021), SPS (Loizou et al., 2021), the optimisation methods used in Vaswani et al. (2019b), SGD with a constant learning rate ($SGD_{Const}$) and SGD with a step learning rate schedule ($SGD_{Step}$). $SGD_{Step}$ benefits from a manually tuned learning rate schedule developed by He et al. (2016) while all other methods have at most a single step size or maximum step size hyperparameter that is cross validated as powers of ten. Hence, $SGD_{Step}$ requires far more tuning and does not provide a fair comparison. However, it is included for completeness. For all optimisation algorithms the problem regularisation hyperparameter is selected from $\lambda \in \{1^{-3}, 1^{-4}, 1^{-5}, 0\}$. ALI-G uses constraint based regularisation, (see section 3); $r$ was selected from $r \in \{50, 100, 200, \infty\}$. All other hyperparameters are left at their default values. $SGD_{step}$ we use the learning rate schedules detailed in He et al. (2016). See Appendix E for more details of hyperparameters used. We reuse the schedule proposed for the CIFAR data sets for SVHN and Tiny ImageNet, reducing the

learning rate by a factor of 10 both half way and three quarters through training. A fixed batch size of 128 and an epoch budget of 200 are used for all experiments. As is common for deep learning experiments we accelerate SGD, ALIG and ALI-G+ with a Nesterov momentum of 0.9. $SLS_{Polyak}$, Adam and Adabound also include momentum like terms which we leave at their default settings. We performed 3 runs for all experiments and report the average performance.

| | SVHN | Cifar10 | | Cifar100 | | Tiny ImageNet | | ImageNet |
| --- | --- | --- | --- | --- | --- | --- | --- | --- |
| | | Test Acc (%) | | | | Val Acc (%) | | |
| Model | | | | Small ResNet | | | | ResNet18 |
| Data Aug | No | No | Yes | No | Yes | No | Yes | Yes |
| $SGD_{Step}$ | 95.4 | 84.1 | 87.6 | 51.0 | 59.6 | 39.8 | 43.2 | 71.1 |
| $SGD_{Const}$ | 94.2 | 80.6 | 87.0 | 49.0 | 57.3 | 36.6 | 41.3 | 57.5 |
| $ALIG$ | 93.8 | 80.6 | 86.2 | 47.5 | 57.9 | 35.6 | 41.8 | 63.7 |
| $SPS$ | 93.9 | 81.1 | 86.8 | 43.7 | 53.1 | 20.5 | 23.8 | 63.9 |
| $Adabound$ | 93.1 | 75.6 | 85.2 | 44.0 | 55.4 | 34.2 | 40.1 | 62.9 |
| $Adam$ | 94.0 | 79.7 | 86.0 | 48.1 | 56.2 | 35.9 | 41.3 | 62.6 |
| $AdamP$ | 93.8 | 79.9 | 85.9 | 47.6 | 58.2 | 36.3 | 41.7 | 63.5 |
| $Coin$ | 92.1 | 75.5 | 84.1 | 42.4 | 54.0 | 31.0 | 36.2 | 61.5 |
| $SLS_{Armijo}$ | 93.0 | 81.5 | 85.7 | 31.6 | 42.0 | 11.2 | 11.1 | 63.2 |
| $SLS_{Goldstein}$ | 92.3 | 78.3 | 86.4 | 45.5 | 57.2 | 33.0 | 40.4 | 62.6 |
| $SLS_{Polyak}$ | 93.5 | 79.8 | 85.9 | 43.6 | 54.0 | 31.3 | 38.3 | 62.7 |
| $PAL$ | 94.2 | 81.5 | 86.7 | 39.8 | 57.0 | 35.3 | 40.8 | 63.6 |
| ALI-G+ ($K = 5$) | **95.5** | **85.0** | **87.2** | **56.1** | **59.4** | **39.8** | **42.6** | **67.8** |

Table 1: *Accuracies of optimisation methods on a selection of standard image classification data sets. The model and dataset combinations have been chosen to include both interpolating and non-interpolating tasks. The standard deviation of the accuracy was at most 0.3 for ALI-G+ . $SGD_{step}$ is the only method to benefit from a manually designed step size schedule. All other methods have at most one fixed step size hyperparameter. On these tasks ALI-G+ outperforms all other single hyperparameter methods, often by a large margin.*

**Results.** The accuracy of the best performing model for each optimisation method is shown in Table 5. On the tasks considered, ALI-G+ does at least as well as all other line search and adaptive gradient methods. On many tasks it outperforms all other methods by a significant margin. The main exception being on CIFAR10 with data augmentation where $SGD_{Const}$ produced similar test accuracy. The dominant performance of ALI-G+ shows the lack of strong algorithms for non-interpolating settings. The performance benefit of ALI-G+ is most notable when the interpolation property is far from satisfied or when data augmentation is not used. For example, on the challenging Tiny Imagenet data set ALI-G+ produces validation accuracy 4% higher than the next best. Empirically we observe that ALI-G+ is almost twice as fast as the line search methods, significantly faster than Adam, and has a run time within a couple of percent of SGD. Typical training curves for ALI-G+ are shown in Figure 4. We present results for ALI-G+ with different values of $K$ and a global step size in Appendix D

### 5.3 Large Image Classification Experiments

**Setting.** The ImageNet data set (Deng et al., 2009) contains 1.2M large RGB images of various sizes split over 1000 classes. For our experiments we use the following data augmentation. All images are normalised per channel, randomly cropped to 224x224 pixels and horizontal flips are applied with probability 0.5. For validation a centre crop is used and no flips are performed. For ImageNet the ground truth labels are not freely available so we report validation scores instead. We train a ResNet18 containing 11.7M parameters (He et al., 2016). Due to the large number of images and data augmentation the interpolation assumption

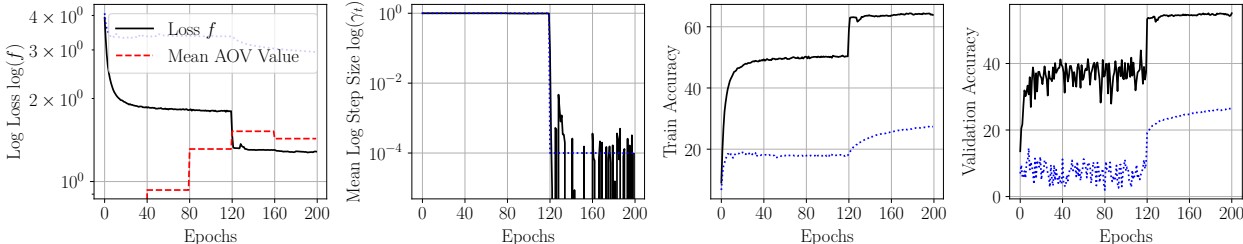

Figure 4: The black solid lines show curves produced by training a small ResNet on CIFAR100 using the ALI-G+ optimiser. Here $\eta = 1.0$, $\lambda = 0.001$ and no data augmentation was used. The AOVs are updated every 40 epochs. Until epoch 120 the mean loss is significantly higher than the mean AOV and thus the maximum step size $\eta$ is used for the majority of updates. The blue dotted curves show SGD with exactly the same hyperparameters as ALI-G+ except using a learning rate schedule this mimics ALIG+ but with the average step size used for all batches. The large difference in performance between these methods demonstrates the superiority of using a step size tailored to each batch. Appendix C we provide additional training curves for ALI-G+ in a variety of settings.

does not hold. We opted to not use a larger model to save on computation requirements and to ensure interpolation was not approximately satisfied.

**Method.** Due to computational constraints for all methods we reuse the best hyperparameters from Tiny ImageNet for Imagenet. However, the batch size is increased to 256 and the epoch budget is reduced to 90. For $SGD_{step}$ we use the learning rate schedule described in He et al. (2016). Note, a different learning rate schedule is suggested in this setting, again highlighting the weakness of using $SGD_{step}$ where a good learning rate schedule is not known in advance.

**Results.** The validation accuracy of each optimisation method is shown in the last column of Table 5. On this task, ALI-G+ outperforms all line search and adaptive gradient methods by at least 3.5%. Additionally, ALI-G+ was significantly quicker to train than two of the next best performing methods. SLS$_{Armijo}$ and PAL took 20 and 12 hours longer to train than ALI-G+ , respectively. These results show the advantage of ALI-G+ for training on large data sets over comparable techniques, especially line search methods.

**Runtime and Memory Analyses.** In Table 2 we include the run time for training a ResNet18 split across two NVIDIA TITAN XP GPUs on ImageNet. ALI-G+'s run time is comparable to adaptive gradient methods and faster than line search approaches. During training ALI-G+ requires that an additional $3|\mathcal{Z}|$ floats are stored where $|\mathcal{Z}|$ is the size of the training set. This extra memory requirement may seem large, however, in many scenarios $3|\mathcal{Z}|$ is orders of magnitude times smaller than the number of parameters of the model $d$. For a concrete example, when training the ResNet18 on ImageNet, $3|\mathcal{Z}| \approx 3.8M$, however, ResNet18 requires 11.7M floats for the parameters alone, at least 11.7M for gradients and then roughly the same again for storing activations and their gradients. Thus, the extra memory requirement of ALI-G+ is negligible.

| Optimiser | SGD | ALI-G | SPS | Adabound | Adam | AdamP | Coin | SLS | PAL | ALI-G+ |
|-----------|-----|-------|-----|----------|------|-------|------|-----|-----|--------|
| Time (h)  | 32  | 34    | 33  | 33       | 35   | 35    | 34   | 53  | 47  | 35     |

Table 2: *Wall clock time for training ResNet18 on ImageNet with various single hyperparameter methods. Here ALI-G+ is as fast as adaptive gradient methods.*

### 5.4 NLP Experiments

**Setting.** For NLP Experiments we consider two tasks. The first is binary classification of reviews on the IMDB data set using a bi-directional LSTM. The second is the training of a Recurrent Neural Network (RNN) for character-level language modelling on the Tolstoi War and Peace data set which forms part of the DeepOBS benchmark (Schneider et al., 2019). The bi-directional LSTM has 1 layer and the RNN has 2 layers. Both models have 128 hidden units per layer. By selecting these models the interpolation property is satisfied on the IMDB data set but not on the Tolstoi data set.

**Method.** We compare ALI-G+ against the majority of the algorithms used in section 5.2. However, we use a slightly modified cross validation scheme; each optimiser's step size or maximum step size hyperparameter is again cross validated as powers of ten. Following (Sivaprasad et al., 2020) the weight decay amount $\lambda$ was selected from $\{0.01, 0.001\}$ for the IMDB classification task, and was not applied to biases. For this task a batch size of 128 and an epoch budget of 100 was used. In contrast, no regularisation, a batch size of 50 and an epoch budget of 150 was used for Tolstoi character prediction.

**Results.** On the easy IMDB review classification task a large number of the optimisation methods achieved close to zero training loss and similar accuracies. The best performing of these were Adam and ALI-G+ which attained a test accuracy of 87.9% and 88.0%, respectively. For the harder character prediction task using the Tolstoi data set ALI-G+ was the best performing algorithm by over 1%. The full results are shown in Table 3. These results reinforce i) that ALI-G+ consistently achieves highly competitive results in a wide range of settings; and ii) in the non-interpolating setting ALI-G+ is particularly effective.

| Data Set | Model | $SGD_{const}$ | Adabound | Adam | Coin | $SLS_{Armijo}$ | $SLS_{Goldstein}$ | $SLS_{Polyak}$ | PAL | ALI-G+ (K=5) | ALI-G+ (K=10) |
|---|---|---|---|---|---|---|---|---|---|---|---|
| IMDB | LSTM | 87.5 | 82.7 | 87.9 | 87.6 | 73.4 | 78.6 | 67.1 | 85.1 | **88.0** | 87.7 |
| Tolstoi | RNN | 49.4 | 41.7 | 57.9 | 56.9 | 30.0 | 39.0 | 31.3 | 52.9 | **59.7** | 59.4 |

Table 3: *Test accuracy of single hyperparameter optimisation methods on NLP data sets. For both data sets ALI-G+ is the best performing task with Adam in a close second. However, on the relatively easy IMDB review classification task a large number of the optimisation methods achieved close to zero training loss and similar test accuracy.*

## 6 Discussion

We have introduced ALI-G+, an optimisation algorithm designed for settings where interpolation does not hold. We have demonstrated the effectiveness of ALI-G+ on many standard data sets outperforming a wide range of modern neural network optimisation techniques both in run time and generalisation performance. We now briefly discuss four directions for future work. The first would be to characterise the conditions where ALI-G+ offers provable convergence. The second idea is a modification of ALI-G+ when data augmentation is used. Here, one could make use of a convenient distribution, to model the current spread of $\ell_z(\bar{w})$ given by the data augmentation. When updating the AOVs one could then use a lower confidence bound on this distribution. The Third direction for future work is the application of ALI-G+ to distillation where the teacher network could be used to generate AOVs for the student. Finally, it maybe possible to combine ALI-G+ with the ideas presented in Paren et al. (2022) to increase it's robustness to its hyperparameters.

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

## A    Mini Batch Step-Size

In this section we detail the step size formulae (6) when a applied to a mini-batch. We define the set of indexes $z_t$ of the example selected with in the batch at time $t$ as $\mathcal{B}_t$. With this notation $|\mathcal{B}_t|$ denotes the batch size. The loss values, AOV's and gradients are simply averaged over the batch.

$$\gamma_t \triangleq \max\left\{\min\left\{\eta, \frac{\mathbb{E}_{z_t \in \mathcal{B}_t}[\ell_{z_t}(\boldsymbol{w}_t) - \tilde{\ell}_z^k]}{\|\mathbb{E}_{z_t \in \mathcal{B}_t}[\nabla_{\boldsymbol{w}}\ell_{z_t}]\|^2}\right\}, 0\right\}. \tag{8}$$

## B    Unsuccessful Approaches

In this appendix we briefly introduce so unsuccessful ideas for both the Algorithms 1 and 2.

**Parameter Updates**    A idea we experiments with for Algorithms 1 was defining and augmented loss function $\ell_z'$ as the point-wise maximum of the $z^{th}$ loss function and its relevant AOV, specifically:

$$\ell_z'(\boldsymbol{w}) \triangleq \max\left\{\ell_{z_t}(\boldsymbol{w}_t), \tilde{\ell}_z^k\right\}. \tag{9}$$

This augmented loss function would then be used in the following parameter update:

$$\boldsymbol{w}_{t+1} = \boldsymbol{w}_t - \gamma_t \mathbb{E}_{z_t \in \mathcal{B}_t}[\nabla_{\boldsymbol{w}}\ell_{z_t}'(\boldsymbol{w}_t)], \tag{10}$$

$$\gamma_t \triangleq \max\left\{\min\left\{\eta, \frac{\mathbb{E}_{z_t \in \mathcal{B}_t}[\ell_{z_t}'(\boldsymbol{w}) - \tilde{\ell}_z^k]}{\|\mathbb{E}_{z_t \in \mathcal{B}_t}[\nabla_{\boldsymbol{w}}\ell_{z_t}']\|^2}\right\}, 0\right\}, \tag{11}$$

where $\mathcal{B}_t$ is the set of indexes $z_t$ of the example selected with in the batch at time $t$. This formulation excluded loss functions $\ell_z$ that had reached their AOV both in the step size calculation and the descent direction. This had the effect to focus on examples that had not yet reached their AOV, while at first this might sound desirable, in the non-interpolation setting it is counter productive. In this setting it is given we can't achieve zero loss on all sample simultaneously, hence focusing on the hardest samples are detrimental when trying to minimise the mean loss.

**AOV Updates** The AOV increase in lines 9-10 in Algorithm 2 was inspired by Hazan & Kakade (2022) and we did not try alternate schemes. For the AOV decrease 6-7 in Algorithm 2 we initially tried back tracking to the previous AOV that had been reached rather than half way. We found this worked slightly worse in practice as it resulted in the AOVs oscillating more.

## C  Additional Plots

In this section we provide a variety of training curves produced by ALI-G+ in a number of settings. We start by showing ALI-G+ 's performance on the SVHN data set where interpolation holds, see Figure 5. The next few plots detail runs of ALI-G+ on the slightly more challenging CIFAR100 data set. In Figure 6, in contrast to 4 we show the behaviour of ALI-G+ both without data augmentation. In Figure 7 we show training curves when $K = 10$ to highlight why $K = 5$ is preferred. Finally, we show the behaviour when using ALI-G+ to train a ResNet18 (He et al., 2016) on the ImageNet data set.

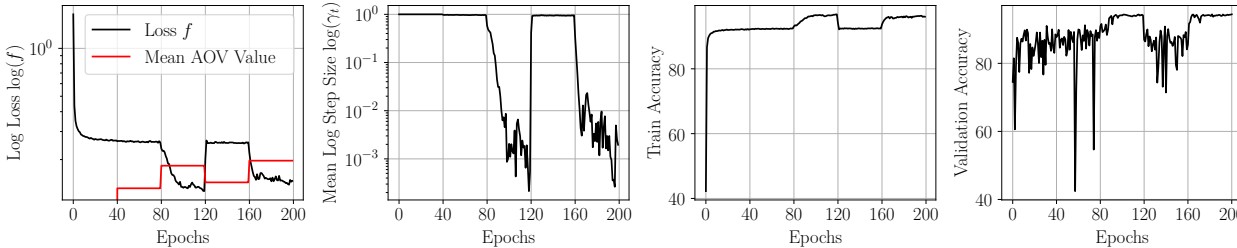

Figure 5: Curves produced by training a small ResNet on the SVHN data set (Netzer et al., 2011) with the ALI-G+ optimiser. These results were produced with $\eta = 1.0, \lambda = 10^{-3}$. No data augmentation was used as is standard for SVHN. The AOVs were updated every 40 epochs. The maximum step size $\eta$ is selected for the first 80 epochs. At epoch 80 the AOVs are increased to a point where the mean step size decreases. This was followed by a sharp decrease in loss value over the next 20 epochs. This, in turn, results in the mean step size dropping further and becoming zero for many batches. At epoch 120 the mean AOV value was significantly higher than the mean loss value $\ell_z(\bar{\boldsymbol{w}}_k)$ resulting in the majority of AOVs being decreased in value during the update. The updated AOV values resulted in the maximum step size being selected again for most batches. This causes the loss to increase sharply. Finally, at epoch 160 the AOVs are increased again resulting in a similar behaviour to a at epoch 80.

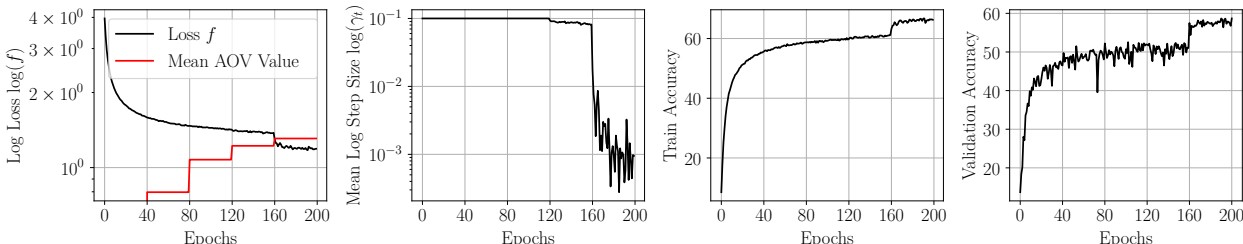

Figure 6: Curves produced by training a small ResNet on CIFAR100 data set (Krizhevsky, 2009) with data augmentation with the ALI-G+ optimiser. These results were produced with $\eta = 0.1, \lambda = 10^{-3}$. The AOVs are updated every 40 epochs. The maximum step size $\eta$ is selected for the majority of batches during the first 120 epochs. For the remaining 140 epochs the step size was tailored to each batch.

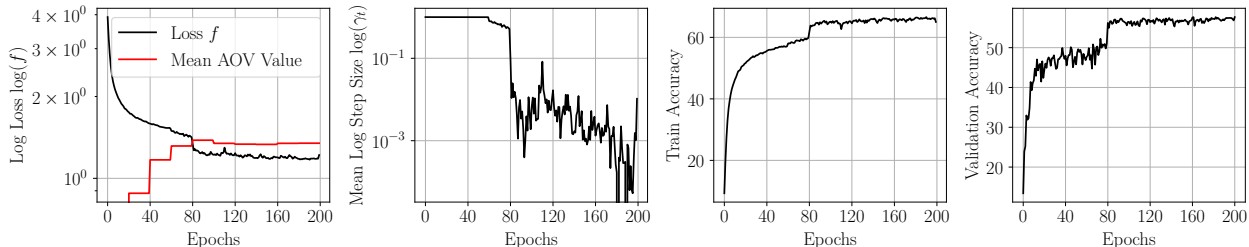

Figure 7: Curves produced by training a small ResNet on CIFAR100 data set (Krizhevsky, 2009), again with data augmentation with the ALI-G+ optimiser, however, we set $(K = 10)$. These results were produced with $\eta = 1.0, \lambda = 10^{-4}$. The AOVs are updated every 20 epochs. The maximum step size $\eta$ is selected for the majority of batches during the first 60 epochs. For the last 140 epochs the step size was tailored to each batch. However, the accuracy does not improve significantly during the last half of training. Due to the rapid AOV updates the mean AOV stabilises at a suboptimally high value. This results in small step sizes being used on average and thus little progress is made for the remaining of the training period. This results in $K = 10$ achieving slightly worse accuracy than when $K = 5$. Empirically we find $k = 5$ provides good results for all settings considered.

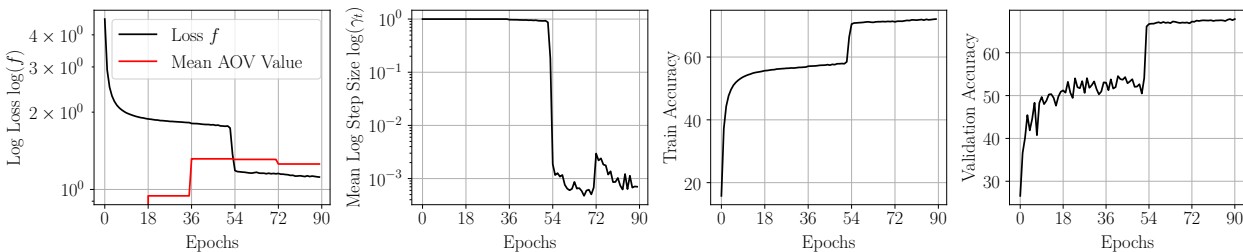

Figure 8: Curves produced by training a ResNet18 on the ImageNet data set (Deng et al., 2009) with data augmentation with the ALI-G+ optimiser. These results were produced with $\eta = 1.0, \lambda = 10^{-4}$. The AOVs are updated every 18 epochs. The maximum step size $\eta$ is selected for the majority of batches during the first 36 Epochs. For the remaining 54 epochs ALI-G+ tailors the step size to each batch.

## D    Additional Results

In this appendix we provide results for modified versions of ALI-G+ applied to many of the image classification tasks detailed in Section 5.2 of the main paper. We state result for 3 additional different values of

$K$ and for two version of ALI-G+ that use a global step size instead of a per sample step size. These steps sizes are detailed below. The global AOV $\tilde{f}^k$ that aims to approximate $f_\star$ is updated using the equations in Algorithm 2 with the the point wises estimates $\ell_{z_t}(\bar{\boldsymbol{w}})$ replaced with their global versions $\tilde{f}(\bar{\boldsymbol{w}})$.

**Global ALI-G+ variant 1 (GALIG1+)**

$$\gamma_t \triangleq \max\left\{\min\left\{\eta, \frac{|\mathcal{B}|\sum_{z\in\mathcal{B}}[f(\boldsymbol{w}_t) - \tilde{f}^k]}{\|\sum_{z\in\mathcal{B}}[\nabla_{\boldsymbol{w}}\ell_{z_t}]\|^2}\right\}, 0\right\}. \tag{12}$$

**Global ALI-G+ variant 2 (GALIG2+)**

$$\gamma_t \triangleq \max\left\{\min\left\{\eta, \frac{|\mathcal{B}|\sum_{z\in\mathcal{B}}[\ell_{z_t}(\boldsymbol{w}_t) - \tilde{f}^k]}{\|\sum_{z\in\mathcal{B}}[\nabla_{\boldsymbol{w}}\ell_{z_t}]\|^2}\right\}, 0\right\}. \tag{13}$$

**Results**  Table 4 details the results of these experiments. ALI-G+ performs best for $K \in \{5, 10\}$, with the accuracy's falling away as $K$ is increase or decreased outside this range. GALIG1+ performs similar to ALI-G+ , which indicates that using a single global AOV could be sufficient in these settings. This could be a promising direction for future work that would reduce the memory foot print. GALIG2+ performs slightly worse than both ALI-G+ and GALIG1+. This suggests in practice mixing batch-wise and global estimates of the loss in the step size should be avoided, however further investigation is needed.

| | SVHN | Cifar10 | | Cifar100 | | Tiny ImageNet | | ImageNet |
|---|---|---|---|---|---|---|---|---|
| | | Test Acc (%) | | | | Val Acc (%) | | |
| Model | | | | Small ResNet | | | | ResNet18 |
| Data Aug | No | No | Yes | No | Yes | No | Yes | Yes |
| *ALIG* $(K \approx 1)$ | 93.7 | 80.8 | 86.2 | 47.5 | 57.9 | 35.6 | 41.8 | 63.7 |
| ALI-G+ $(K = 3)$ | 95.4 | 84.8 | 86.5 | 54.6 | 57.4 | 36.1 | 41.4 | - |
| ALI-G+ $(K = 5)$ | **95.5** | **85.0** | **87.2** | 56.1 | **59.4** | 39.8 | **42.6** | **67.8** |
| ALI-G+ $(K = 10)$ | 95.0 | **85.0** | 86.8 | **56.6** | 58.0 | **39.9** | 42.3 | 67.1 |
| ALI-G+ $(K = 20)$ | 94.8 | 83.8 | 85.9 | 54.5 | 56.0 | 38.9 | 40.8 | 65.1 |
| ALI-G+ $(K = 5)$ | **95.5** | **85.0** | 87.2 | 56.1 | **59.4** | 39.8 | **42.6** | - |
| GALIG1+ $(K = 5)$ | **95.5** | 84.4 | **87.3** | **56.2** | **59.4** | **40.4** | **42.6** | - |
| GALIG2+ $(K = 5)$ | 94.8 | 84.5 | 86.3 | 56.0 | 56.9 | 35.5 | 41.3 | - |

Table 4: *Accuracies for ALI-G+ with $K = 5$, $K = 10$ and $K = 20$ on a selection of standard image classification data sets. ALI-G+ with $K = 10$ offers comparable results to $K = 5$, however, when $K$ is increased to $K = 20$ the performance becomes noticeably worse. We additionally show two results for ALI-G+ with a global step size.*

# E  Hyperparameters

In this appendix we give great detail on the hyperparameters used in Section 5.2.

| Optimiser | Step-size | Regularisation |
|---|---|---|
| $SGD_{Step}$ | $\eta_0 \in \{0.1, 0.01\}$ | wd $\in \{0, 1^{-5}, 1^{-4}, 1^{-3}\}$ |
| $SGD_{Const}$ | $\eta \in \{0.1, 0.01\}$ | wd $\in \{0, 1^{-5}, 1^{-4}, 1^{-3}\}$ |
| $ALIG$ | $\eta_{max} \in \{0.1, 0.01\}$ | r $\in \{0, 50, 100, 200\}$ |
| $SPS$ | $\eta_{max} \in \{1, 10, 100\}$ | $\ell_2 \in \{0, 1^{-5}, 1^{-4}, 1^{-3}\}$ |
| $Adabound$ | $\eta \in \{0.01, 0.001\}$ | wd $\in \{0, 1^{-5}, 1^{-4}, 1^{-3}\}$ |
| $Adam$ | $\eta \in \{0.01, 0.001\}$ | wd $\in \{0, 1^{-5}, 1^{-4}, 1^{-3}\}$ |
| $AdamP$ | $\eta \in \{0.01, 0.001\}$ | wd $\in \{0, 1^{-5}, 1^{-4}, 1^{-3}\}$ |
| $Coin$ | N/A | $\ell_2 \in \{0, 1^{-5}, 1^{-4}, 1^{-3}\}$ |
| $SLS_{Armijo}$ | N/A | $\ell_2 \in \{0, 1^{-5}, 1^{-4}, 1^{-3}\}$ |
| $SLS_{Goldstein}$ | $\eta_{max} \in \{1, 10\}$ | $\ell_2 \in \{0, 1^{-5}, 1^{-4}, 1^{-3}\}$ |
| $SLS_{Polyak}$ | N/A | $\ell_2 \in \{0, 1^{-5}, 1^{-4}, 1^{-3}\}$ |
| $PAL$ | $\eta_{max} \in \{1, 10\}$ | $\ell_2 \in \{0, 1^{-5}, 1^{-4}, 1^{-3}\}$ |
| ALI-G+ | $\eta_{max} \in \{0.1, 0.01\}$ | wd $\in \{0, 1^{-5}, 1^{-4}, 1^{-3}\}$ |

Table 5: *Hyperparameters cross-validated in the experiements in Section 5.2. All other hyperparameters were left at the default values as specified by the official authors implementation (PAL, SLS, SPS, Coin, AdamP, ALIG, Adabound) or the PyTorch implentaiton (SGD, Adam).*

