# OpenReview forum: "Faking Interpolation Until You Make It"
_TMLR — Accepted by TMLR_

### Review · Reviewer_DvfA · 2022-07-29

**Summary Of Contributions:**

The paper’s main contribution is a new optimization algorithm called ALIG+. This method is a modification of the original ALI-G algorithm for interpolating problems. ALI-G requires that the optimal loss value is known a priori, whereas ALIG+ uses a heuristic to estimate the optimal loss value. While no convergence guarantees are provided, the authors do not claim this as a contribution. Instead, they focus on large scale empirical evaluation of the optimizer against previous methods on a variety of tasks. The authors claim that ALIG+ has similar walk-clock time to SGD and provides competitive generalization results.

**Broader Impact Concerns:**

N/A.

**Requested Changes:**

- The discussion of SPS from Loizou et al., 2021 is confusing. It's not clear to me exactly where interpolation is required in that work. Indeed there is Assumption 2.1, but this is not interpolation. Can the authors comment on this?
- The accuracies obtained on Table 1 are generally quite low. For example, less than 90% on CIFAR10 with data augmentation is very far from SoTA. I understand that the model size is limited to avoid interpolation, but is there any other reason for this?
- This is particularly concerning since Loizou et al., 2021 report accuracies of >92% on CIFAR10 and >72% on CIFAR100 using SLS and SPS.
- How is the bolding chosen in Table 1? Currently it's very misleading. The entire ALIG+ K=5 row is bold, but frequently it does not actually achieve the best accuracy on the task. The authors also admit a standard deviation of ~0.3, which would mean several of the confidence intervals would be overlapping.
- I'm confused about the purpose of Fig. 3. Why is the correct comparison to use the same hyperparameters for SGD that were tuned for ALIG+?

**Strengths And Weaknesses:**

Strengths
- There are a relatively large number of experiments on different problems and data modalities.
- The proposed algorithm is efficient in terms of its time and memory overhead compared to standard SGD.

Weaknesses
- No convergence guarantees are provided.
- It's unclear how well the baselines were tuned, since the authors provide their own results with previous methods rather than comparing against published results.
- While ALIG+ does not seem worse than previous methods, there aren't many cases where it is obviously better.

---

> ### Author Response · Authors · 2022-08-07
> **Responce to suggested changes**
>
> Many thanks for your review and comments. Here we aim to respond to your suggested changes.
>
> * The discussion of SPS from Loizou et al., 2021 is confusing. It's not clear to me exactly where interpolation is required in that work. Indeed there is Assumption 2.1, but this is not interpolation. Can the authors comment on this?
>
> To our understanding, the authors of SPS give approaches to obtain fi* in a number of settings, non-interpolating deep learning problems, were not one of them. Moreover, in our experiments SPS did not perform well in this setting. If our understanding is incorrect please could you point us to the section where this is explained. We are happy to change the wording if you think we have misrepresented where SPS can be easily applied.
>
> * The accuracies obtained on Table 1 are generally quite low. For example, less than 90% on CIFAR10 with data augmentation is very far from SoTA. I understand that the model size is limited to avoid interpolation, but is there any other reason for this?
>
> The other major reason was computational constrains. We would have liked investigate larger non-interpolating models trained on even larger data sets, such as ImageNet. However, this was outside what was reasonable with our access to compute. We agree we do not come close the SoTA in term of absolute performance when model size is ignored. However, we think the training of small networks for low compute applications is still a research area of importance.
>
> * This is particularly concerning since Loizou et al., 2021 report accuracies of >92% on CIFAR10 and >72% on CIFAR100 using SLS and SPS.
>
> The ResNet34 used by SPS has 65M parameters, the model used in this paper has approximately 0.1% of that (0.09M). Hence we don’t think it is concerning there is a big difference in performance. The training of large interpolating models is not a setting where we recommend ALI-G+. Instead as stated in the paper we would suggest vanilla ALI-G be used, which leads to a performance of 93.5% on CIFAR10 and 76% on CIFAR100 for more details see “Comment on Stochastic Polyak Step-Size: Performance of ALI-G” Berrada et al. 2021.
>
> * How is the bolding chosen in Table 1? Currently it's very misleading. The entire ALIG+ K=5 row is bold, but frequently it does not actually achieve the best accuracy on the task. The authors also admit a standard deviation of ~0.3, which would mean several of the confidence intervals would be overlapping.
>
> Thanks for raising this bolding issue. The reason is we did not consider K=10 and K=20 in the comparison. These results are just included for completeness, ALI-G+ has k fixed at 5, as we found this gave good results over a wide range of problems. We will amend the formatting and wording to make this less misleading. Regarding overlapping confidence intervals, we will adjust the text in 5.2 to make sure our claim’s reflect this.
>
> * I'm confused about the purpose of Fig. 3. Why is the correct comparison to use the same hyperparameters for SGD that were tuned for ALIG+?
>
> The purpose of Fig. 3. is to indicate that the real performance benefit of ALI-G+ comes from using a step-size tailored to each batch, not the mean learning rate schedule it recovers. Using the mean step size on each batch, which is equivalent  to SGD, does not produce good results. We will adjust the wording to make sure this is clear.

---

### Review · Reviewer_JZit · 2022-08-09

**Summary Of Contributions:**

This work proposes a variant of a certain adaptive step-size algorithm (ALI-G), which is applicable to non-interpolating settings. The ALI-G optimizer from prior work was a variant of Polyak step-size / Newton-Raphson, and thus only applied to settings where the optimal loss is 0 (or is known).
This paper extends the optimizer by introducing a heuristic estimation of the optimal-loss per-sample, which enables stochastic optimization for non-interpolating settings (e.g. with large data sizes or small model sizes). They conduct fairly extensive empirical studies showing that the new optimizer is competitive in modern settings.

I recommend acceptance to TMLR.

**Requested Changes:**

Suggestions which do not affect the score:
- I wonder if the learned pointwise losses are related to the "difficulty" of individual samples. I expect they will be; it could be interesting to engage with the "example difficulty" literature.
- I suggest clarifying how ALI-G handles mini-batching. (eg: is there a "batch AOV" that's the average of the sample-AOVs?)
- It would be informative to plot the "effective step size" \gamma_t chosen by ALIG+, as a function of time T. This would clarify how ALIG+'s adaptive step size schedule differs from other standard choices (step-wise decay, cosine decay, etc).


**Strengths And Weaknesses:**

Strengths:
- Well-written
- Preliminaries that introduces reader to context and prior work.
- Good motivation: non-interpolating methods are often used in large-data settings
- Interesting method: estimating pointwise losses online, and using them for optimization, is an idea that has not been explored thoroughly in prior work
- Extensive experiments on a wide variety of settings (including both DNN and non-DNN)

Weaknesses:
- Some of the experiments (specifically the vision-tasks) appear to somewhat over-claim the performance of the new method. Specifically, in Table 1, ALI-G+ is claimed to be the best "single hyperparameter" method. However, this restriction to "single hyperparameter" methods is both artificial (since even ALI-G+ uses L2 regularization, some choice of batch-size, choice of epochs, etc), and not thoroughly explored. For example, SGD with cosine LR decay is "single hyperparameter" in the terminology of this paper, and is very competitive for CIFAR – but not included in Table 1.
- Figure 3: Showing the blue dotted line as SGD with the "average step-size of ALIG+" seems a bit unfair. Since ALIG+ actually uses a *constant* stepsize until Epoch ~120, it would be more fair to compare against SGD with this constant stepsize.
- The "AOV update scheme" is somewhat ad-hoc, and there is not much discussion into why this rule was chosen, and the effect of alternate update rules.
- The experimental results show top-line accuracies, but there is little investigation/ablations into what factors in ALIG+ contribute to its performance. For example, how important is it that we keep track of pointwise losses, instead of a uniform lower-bound for all points?
- There is no code provided (AFAIK). Including code in the final submission, especially for optimizer contribution, will make this submission both more reproducible and more impactful.

---

> ### Author Response · Authors · 2022-08-23
> **Response to highlighted weaknesses and requested changes.**
>
> Thank you for your review.
>
> Some of the experiments (specifically the vision-tasks) appear to somewhat over-claim the performance of the new method. Specifically, in Table 1, ALI-G+ is claimed to be the best "single hyperparameter" method. However, this restriction to "single hyperparameter" methods is both artificial (since even ALI-G+ uses L2 regularization, some choice of batch-size, choice of epochs, etc), and not thoroughly explored.
>
> - SGD with cosine LR decay or SGDR as introduced by Loshchilov and Hutter 2017 has by our count three hyperparameters eta_min, eta_max and a hyperparameter governing the restart frequency. In the SGDR paper the authors even suggest changing these quantities though training. In their experiments they tune all three of these hyperparameters. In contrast, in our experiments we only tune the maximum learning rate of ALI-G+, and the task regularization for all methods. We agree nearly all stochastic first order methods contain other hyperparameters (L2 regularization, some choice of batch-size, choice of epochs, etc) . We did not include them in the total number as they are common to all methods. While we did not investigate changing these hyperparameters we chose what we thought were common values widely used in the literature. We did not have access to computational resources to give a fair comparison against a large number of baseline when investigating a grid search of more than two dimensions. We have included this explanation in the revised version of the paper in order to address the reviewers' concern regarding the claims made in our experiments
>
> Figure 3: Showing the blue dotted line as SGD with the "average step-size of ALIG+" seems a bit unfair.
>
> - The purpose of Fig. 3. is to indicate that the real performance benefit of ALI-G+ comes from using a step-size tailored to each batch, not the mean learning rate schedule it recovers. Using the mean step size on each batch, which is equivalent to SGD, does not produce good results. We have adjusted the wording to make sure this is clear.
>
> The "AOV update scheme" is somewhat ad-hoc, and there is not much discussion into why this rule was chosen, and the effect of alternate update rules.
>
> - Our scheme for increasing the AOVs (algorithm.2 9&10) was based on the non-stochastic scheme detailed in Revisiting the Polyak Step Size 2022 Hazan and Kakade. While the previous version of this paper contained sign errors, this appears to now have been corrected on arxiv as of two weeks ago. We have included a citation of the corrected version in our revised paper. Note that, in contrast to our work, this prior work considers the non-stochastic case. Hence, its direct use is computationally prohibitive in deep learning settings. In the non-stochastic case, the AOV is always a lower bound on f(star), hence there is no need to decrease the AOVs. The scheme for decreasing the AOVs (algorithm.2 6&7) simply selects the AOV to be half-way between the past previous AOV that was “reached” and the value that wasn’t.
>
> The experimental results show top-line accuracies, but there is little investigation/ablations into what factors in ALIG+ contribute to its performance. For example, how important is it that we keep track of pointwise losses, instead of a uniform lower-bound for all points?
>
> - Thank you for this suggestion we have included two baselines that just use a single global AOV for comparison.
>
> There is no code provided (AFAIK). Including code in the final submission, especially for optimizer contribution, will make this submission both more reproducible and more impactful.
>
> - Code will be available.
>
> Requested Changes:
>
> I wonder if the learned point-wise losses are related to the "difficulty" of individual samples. I expect they will be; it could be interesting to engage with the "example difficulty" literature.
>
> - This is a good observation and yes we took inspiration from the curriculum learning literature. This scheme can be thought of as defining a curriculum for the non-interpolating setting. Rather than first focusing on easy examples, at the beginning all examples are treated equally. After time, examples that are identified as hard are given less importance, and the optimiser does not try to optimise these examples further. As we know we cannot achieve zero loss on all samples simultaneously it makes sense to not focus over the hardest examples.
>
> I suggest clarifying how ALI-G handles mini-batching.
>
> - The loss values, AOV and gradient values are simply averaged over the batch. We have included an explicit mini batch step size formula in the appendix.
>
> It would be informative to plot the "effective step size" \gamma_t chosen by ALIG+, as a function of time T.
>
> - The second panel of figures 4-8 (updated version) provide the effective step size \gamma_t chosen by ALI-G+ as a function of time

---

### Review · Reviewer_LejS · 2022-08-09

**Summary Of Contributions:**

- This paper proposes a novel optimization algorithm that modifies an optimizer designed to be used in the interpolation setting, such that it can be used in the non-interpolation setting. The resulting optimizer is called ALI-G+, as it builds upon ALI-G. In ALI-G’s update rule, there is a term that corresponds to the loss value of the optimal model, which can be set to the known lower bound in the interpolation setting. ALI-G+ uses the same update rule, but aims to estimate the loss value of the optimal model for each sample in the training set, since the loss doesn’t reach its lower bound in the non-interpolation setting.
- The loss values are estimated to approximately track the loss value corresponding to the best model so far.
- Experiments on a variety of different problems ranging from matrix factorisation to image classification, covering both the interpolation and non-interpolation setting, ALI-G+ is shown to consistently perform better than all considered single-hyperparameter optimizers.




**Broader Impact Concerns:**

There doesn’t seem to be any ethical concerns in this paper.

**Requested Changes:**

**Major points**

Section 4 is hard to parse. Important details are not mentioned or highlighted enough. More specifically:
- Is there a different learning rate per sample in a minibatch? If so, the notations in Equation 6 makes this unclear since $\ell_{z_t}$ is subscripted with $z_t$ while $\tilde{l}^k_z$ is subscribed with $z$. If not, how exactly is the learning rate computed?
- If weight decay is applied directly instead of projecting into a constrained set as in ALI-G, doesn’t this change the objective function, and therefore the derivation of the update rule (Equation 3 and 4)?
- How exactly was $\bar{\mathbf{w}}$ determined in Algorithm 1? How was $\mathbf{\ell}(\bar{\mathbf{w}})$ computed? It seems expensive to compute the losses for all samples.
- In Algorithm 2, why is the “has been reached” case (lines 6-7) different from the “hasn’t been reached” case (lines 9-10)? If we want to decrease the loss values, why are we averaging with the previous loss values? What if the previous loss value is higher? What exactly is the purpose of line 7 as opposed to line 10? Have the authors tried different variants of the update rule for Algorithm 2? It would be informative to include those results in the experiments.

Authors should include the exact ranges and increments of the tuned hyperparameters for the different optimizers in a table.

It doesn’t seem too much of an overhead to perform 3 runs for the baseline methods. I think it’s only fair to do this when it’s done for ALI-G+.


**Minor points**

- The experiments use unrealistically small models. For example, ResNet-18 seems too small to be used for ImageNet. It would be nice to see bigger models like ResNet-50 or Vision Transformers.
- In the introduction: “One simply modifies each loss to be the point-wise maximum of the loss function and its value at the optimal point.” Did the authors mean point-wise minimum?
- Page 2 when the authors cite Adam, the citation is wrong (should be Kingma & Ba instead of Kingma & Welling).
- Page 3: “In order for the interpolation assumption to hold for (P)...” is this a typo? What is P?
- Page 3: “hence the numerator of the fraction in (3)” -> should be (4)
- Page 6: “update these methods use identical descent diration and …” typo in diration -> direction.
- I feel like the whole Justification section in page 6 could be removed, since the authors did not construct a worse case bound.
- In section 5.1, ALI-G isn’t included in one of the baselines. Why?
- Figure 2: I think it makes sense to make the range of the y-axis smaller for the second and fourth plot.
- Page 7: “For the Tiny ImageNet dataset the ground truth labels are not freely available” do the authors mean the test set is not available?
- Page 8: “For SGD we use the learning rate schedules…” -> “For $\text{SGD}_\text{step}$.
- Why was weight decay not used for the biases for the IMDB experiments, and why was regularization not used for the Tolstoi experiments?
- Table 3 second column should be labeled Tolstoi instead of IMDB


**Strengths And Weaknesses:**

**Strengths**

- The idea of using ALI-G and estimating the optimal loss values is novel.
- Experiments are extensive. It includes many tasks, not just image classification. It also includes many relevant baselines.
- The proposed method yields better performance in terms of generalization ability than many single-hyperparameter optimizers, which includes SGD with momentum, Adam, and SLS and PAL (well-performing line search methods).
- Computational overhead of ALI-G+ is slight compared to SGD, and better than adaptive optimizers and line search methods.

**Weaknesses**

- The exact algorithm for approximating the optimal loss values is a bit ad hoc. There must have been other approximation schemes. Why was this specific method chosen?
- The method introduces an additional hyperparameter $K$. Though the authors fixed this to be 5 for all the experiments, it still affects the performance of the method. Was there a reason why the number of times to update the loss values is chosen as a hyperparameter instead of the number of steps to use the fixed loss values for ($\frac{T}{K}$ in this case)? It seems like the latter makes more sense, since the best value for $K$ must depend on the total number of steps $T$.
- ALI-G performs worse than SGD with a tuned learning rate schedule. This means that practitioners who want to achieve the best results with ample budget to tune would still prefer to use an optimizer that can be used with a schedule (gives more flexibility).

---

> ### Author Response · Authors · 2022-08-23
> **Response to the concerns raised**
>
> Thank you for your very thorough review.
>
> Weaknesses :
> The exact algorithm for approximating the optimal loss values is a bit ad hoc. There must have been other approximation schemes. Why was this specific method chosen?
>
> - Our scheme for increasing the AOVs (algorithm.2 9&10) was based on the non-stochastic scheme detailed in "Revisiting the Polyak Step Size" 2022 Hazan and Kakade. While the previous version of this paper contained sign errors, this appears to now have been corrected on arxiv as of weeks ago. We have included a citation of the corrected version in our revised paper. Note that, in contrast to our work, this prior work considers the non-stochastic case. Hence, its direct use is computationally prohibitive in deep learning settings. In the non-stochastic case, f(w) is always a lower bound on f(star),  hence there is no need to decrease the AOVs. The scheme for decreasing the AOVs (Algorithm.2 6&7) simply selects the AOV to be half-way between the previous AOV that was “reached” and the value that wasn’t. We did try simply backtracking to the last AOV that was not reached, which performed slightly worse.
>
> The method introduces an additional hyperparameter K. Though the authors fixed this to be 5 for all the experiments, it still affects the performance of the method. Was there a reason why the number of times to update the loss values is chosen as a hyperparameter instead of the number of steps to use the fixed loss values for (T/K) in this case)? It seems like the latter makes more sense, since the best value for K must depend on the total number of steps T.
>
> - You raise a good point, we could have chosen T/K as a hyperparameter of ALI-G+. However when running deep learning experiments there is always an upper limit on the number of epochs that is viable whether explicit or not. Hence we use this limit to reduce the number of hyperparameters. If one has access to significantly more compute, ALI-G+ can be run till convergence before updating the AOVs, and this process could be repeated any number of times.
>
> ALI-G performs worse than SGD with a tuned learning rate schedule. This means that practitioners who want to achieve the best results with ample budget to tune would still prefer to use an optimizer that can be used with a schedule (gives more flexibility).
>
> - This is true, however ALI-G does not need a large budget to tune. It only has a maximal step size hyperparameter that needs to be adjusted, alongside any other task specific hyperparameters.
>
> Major points
> Section 4 is hard to parse. Important details are not mentioned or highlighted enough.
>
> - The loss values, AOV and gradient values are simply averaged over the batch. We have included an explicit mini-batch step size formula in the Appendix.
>
> If weight decay is applied directly instead of projecting into a constrained set as in ALI-G, doesn’t this change the objective function, and therefore the derivation of the update rule (Equation 3 and 4)?
>
> - We tried modifying Equation 3 and 4, to take account of weight decay in the objective function. However we found empirically this gave worse results. We speculate that this is as it becomes unclear what each AOV would represent and hence how to disentangle the loss and regularization. Our weight decay is similar to that of AdamW that seems to be gaining traction over the original version of Adam.
>
> How exactly was wbar determined in Algorithm 1? How was ℓ(wbar) computed? It seems expensive to compute the losses for all samples.
>
> - As stated in the implementation details section, to save on computation we approximate these quantities. We calculate wbar according to Algorithm 1, line 6, however in practice we approximate f(w_t) = E_z(ℓ(w_last)) in the minimum, where ℓ(w_last) is the last calculated loss value for this sample. Algorithm 2 does not actually need wbar, just ℓ_z(wbar) for all z in Z so we never explicitly calculate wbar. In other words, ℓ_z(wbar) is the loss of the zth sample as calculated in the epoch that had the lowest total loss, ignoring the non-stationarity of w.
>
> In Algorithm 2, why is the “has been reached” case (lines 6-7) different from the “hasn’t been reached” case (lines 9-10)? ...
>
> - Due to the character limit we refer to our first response in this reply.
>
> Authors should include the exact ranges and increments of the tuned hyperparameters for the different optimizers in a table.
>
> - We have added a table in the appendix.
>
> It doesn’t seem too much of an overhead to perform 3 runs for the baseline methods. I think it’s only fair to do this when it’s done for ALI-G+.
>
> - We are in the process of running some repeats and will update Table 1 once these have finished.

---

### Review · Reviewer_XcC6 · 2022-08-13

**Summary Of Contributions:**

The paper introduces ALI-G+, which extends the ALI-G optimizer for cases where the optimal objective value is not assumed to be a certain value (like 0). The key idea is to use existing statistics to approximate the optimal objective value (Algorithm 2), and then apply the same algorithm as ALI-G. Experiment results are demonstrated on several settings such as matrix factorization, image and text classification, and language modeling.

**Requested Changes:**

**Suggestion**: It would be nice to have a toy experiment that shows having an AOV that is higher (and possibly closer to the ground truth) than the guessed one is better. If our AOV is too small, the worst case scenario seems to be using the largest learning rate?

**Critical**: This bring us to another concern, which is about the largest learning rate hyperparameter. I don't think the paper discusses the $\eta$ hyperparameter of ALI-G+ in detail, other than in Figure 3. This seems quite important to all other optimization methods and is a "make-or-break" decision. In fact, Figure 3 seems incorrect, as y axis indicates the log of largest learning rate is $10^0$. Common SGD methods would start with a learning rate of $0.1$ -- why is this not always chosen in ALI-G+?

**Critical**: The experiments are mostly done on very toy settings, and the performances are nowhere near state-of-the-art, or even some common models that can be used (like ResNet50). Even a few years ago, we have achieved over accuracy of 90% on CIFAR-10 and 80% on CIFAR100, and here the results are significantly worse. It is hard to identify whether the superior results in ALI-G+ translate to these settings, and it would be nice to have some supportive evidence. The only one closer to a large model is ResNet18 on ImageNet, where SGD step is already better by a large margin.

**Suggestion**: Section 4 can be difficult to understand, and it would be nice to have a figure showing the intuition behind the heuristic.

**Strengths And Weaknesses:**

Strengths:
- Clear motivations and preliminaries.
- Experiments are performed on various domains, and achieving good results.
- The hyperparameter is very simple (just $K$, the number of updates to the lower bounds, and $\eta$ maximum step size allowed).

Weaknesses:
- The hyperparameter $K$'s role is under-explored. The algorithm would become ALI-G when $K = 1$ (i.e., using the lower bound to run the entire algorithm), yet in most cases ALI-G+ with K=5 has the best performance. Why not consider $K = 2, 3, 4$?
- The results are mostly demonstrated on very small networks. The deepest model in the image classification task is with ResNet18, and these are quite small compared to what are commonly used in practice; as a result, the accuracy numbers are pretty far from state-of-the-art, making it harder to gauge the usefulness of ALI-G+ with larger models.
- There are superior baselines, such as SGD with cosine scheduling or K-FAC that are not compared against. The degree of freedom of cosine scheduling is actually smaller (as it only requires total number of iterations, whereas ALI-G+ needs an additional $K$ with the total number of iterations).
- The method actually does not perform any preconditioning, which seems to be a bit weaker than methods that try to approximate Gauss-Newton.

---

> ### Author Response · Authors · 2022-08-23
> **Responce to weaknesses and requested changes**
>
> Thank you for your review. Below we respond to your suggested weaknesses and requested changes.
>
> Weaknesses:
> The hyperparameter K's role is under-explored. The algorithm would become ALI-G when K=1(i.e., using the lower bound to run the entire algorithm), yet in most cases ALI-G+ with K=5 has the best performance. Why not consider K=2,3,4?
>
> - Standard ALI-G Berrada et al. 2020 is ALI-G+ with K=1, which we include as a baseline. We have included some results for K=3 in Appendix D however some of these experiments are still running. Will will update the paper with the final results before the 26th.
>
> The results are mostly demonstrated on very small networks. The deepest model in the image classification task is with ResNet18, and these are quite small compared to what are commonly used in practice; as a result, the accuracy numbers are pretty far from state-of-the-art, making it harder to gauge the usefulness of ALI-G+ with larger models.
>
> - Due to the character limit please see our response on this topic at the bottom of these reply.
>
> There are superior baselines, such as SGD with cosine scheduling or K-FAC that are not compared against. The degree of freedom of cosine scheduling is actually smaller (as it only requires total number of iterations, whereas ALI-G+ needs an additional Kwith the total number of iterations).
>
> - SGDR with cosine scheduling contains 3 hyper parameters that need tuning, eta_max, eta_min, hyperparameter governing the restart frequency. Loshchilov and Hutter 2017 additionally suggest adjusting these hyperparameters during training further increasing the search space. To use ALI-G+ one simply needs to tune the maximum learning rate as K=5 is fixed for all tasks.
>
> The method actually does not perform any preconditioning, which seems to be a bit weaker than methods that try to approximate Gauss-Newton.
> We believe this to be complementary to the main point of ALI-G+, namely, a simple yet effective scheme for extending ALI-G (itself a method that doesn't perform preconditioning) to non-interpolating settings.
>
>
> Requested Changes:
> Suggestion: It would be nice to have a toy experiment that shows having an AOV that is higher (and possibly closer to the ground truth) than the guessed one is better. If our AOV is too small, the worst case scenario seems to be using the largest learning rate?
>
> - AOVs are always initially set to a known lower bound on the loss; zero for non-negative loss functions. During the iterations, the AOV values are adjusted, thereby potentially lowering the learning rate. We would be grateful if the reviewer could expand on the type of toy experiment they would like us to perform.
>
> Critical: This bring us to another concern, which is about the largest learning rate hyperparameter. I don't think the paper discusses the η hyperparameter of ALI-G+ in detail, other than in Figure 3. This seems quite important to all other optimization methods and is a "make-or-break" decision. In fact, Figure 3 seems incorrect, as y axis indicates the log of largest learning rate is 100. Common SGD methods would start with a learning rate of 0.1 -- why is this not always chosen in ALI-G+?
>
> - Figure 3 is indeed correct. ALI-G+ can automatically reduce the step size when the AOVs suggest it is too large. SGD does not have the ability to do this so it makes sense a more conservative learning rate is optimal.
>
> Critical: The experiments are mostly done on very toy settings, and the performances are nowhere near state-of-the-art, or even some common models that can be used (like ResNet50). ....
>
> - ResNet50 contains 23M trainable parameters; the small ResNet used in the paper has under 0.1% of that (0.09M). Hence we don’t think it is surprising there is a big difference in performance. ResNet50 would be an unsuitable choice of model as it would easily achieve zero training loss on the CIFAR10 and CIFAR100 datasets and we are concerned with setting where interpolation does not hold. in The training of large interpolating models is not a setting where we recommend ALI-G+. Instead as stated in the paper if it is likely one is in the interpolation or approximate interpolation setting we would suggest vanilla ALI-G be used instead. This leads to a performance of 93.5% on CIFAR10 and 76% on CIFAR100 for more details see “Comment on Stochastic Polyak Step-Size: Performance of ALI-G” Berrada et al. 2021. Ideally we would have included results for large models trained on even larger data sets, to ensure none interpolation. However, this was not within the limits of our computational resources. We have tried to perform as thorough a set of experiments as possible and our experimental setup is on par with prior work in this area of machine learning in terms of data sets and baselines considered, for example Vaswani el at 2019, Loizou et al. 2021.

---

### Review · Reviewer_qwHd · 2022-08-19

**Summary Of Contributions:**

The overall motivation is to use a stochastic variation of Polyak step sizes introduced via the use of ALI-G algorithm without prior information on how the optimal loss looks, so that there is no need for designing the LR schedule. ALI-G+ uses ALI-G to iteratively update parameters.


**Requested Changes:**

- Address questions mentioned above in the paper, or through discussion
- Confirm if experiments are representative of results you want to show, particularly:

1. poor training performance on some problems needs to be addressed without explaining around these deficiencies (like reaches 10-4 faster, or validation performance is as good)

2. Some of the assumptions need to be explained (like K value, updating l to halfway point, using default settings for Adam etc., and not choosing other schedulers for SGD)

**Strengths And Weaknesses:**

### Strengths
+ General flow of the paper is good; sections like reintroducing ALI-G is appreciated
+ Empirical evaluation with multiple problem types stands out, and is comprehensive enough for this kind of study
+ Well written in general without many minor errors
+ Overall, for non-interpolating settings, results seem promising

### Weaknesses
- Are there other methods to estimate optimal loss?
- Main comparison is with other single parameter methods; but adaptive multi-parameter methods are as easy to implement in practice and should be included, at least as a separate section/discussion.
- Currently while updating AOVs, if the target loss value has not been reached within the K'th section of training, it is updated to a half-way point. This seems quite arbitrary; I agree that the general motivation is correct - that in either case (in Algo 2), you want loss to continuosly decrease, but why should l^{k+1} be the half-way point and not l^k + some other \eps ? Can you discuss the reasoning behind this
- How fast is "too fast" for trending towards optimal loss. How do you threshold/measure this? As an initial answer, running in the [5,10] range and additional K values work, but is this true for all problems - "ALI-G+ is robust to the choice of K"
- Worst case bound using ALI-G is actually a good idea in this case. Does ALI-G+ perform better always? Can you comment in the justification section above Section 5
- For Figure 1 - ALI-G+ underperforms compared to SLS and PAL for higher rank problems. "However it still provides rapid optimization to at worst 10^-4" seems like a digression to the main conclusion. Even iterations to reach 10^-4 is better in the case of PAL and SLS. Does K affect this? Can you do more updates initially? Of course, this might introduce another parameter
- Again for Figure 2 - saying it approaches the same val loss seems like a way to explain out of "ALI-G+ fails to achieve the same training loss as the line search methods". Are you only testing for validation performance throughout or training performance as well?
- "Adam and Adabound also include momentum like terms which we leave at their default settings." - what are these settings and how may these affect your results?
- Earlier the claim was that ALI-G+ is "robust" to choices of K. Seems like reducing K helps from the results as mentioned in the caption of Table 1. What if you reduce it further to (1,5] ? Abstract/intro can be changed to acknowledge this understanding

---

> ### Author Response · Authors · 2022-08-25
> **Response to questions.**
>
> Many thanks for your review.
>
> Are there other methods to estimate the optimal loss?
>
> - In some settings there are approaches to estimate the true optimal loss per sample by some alternative method, Loizou et al. 2021 highlights some of these methods. However in this work we focus on the general case where this is not possible so we estimate these quantities online. It is likely there are other online heuristics aside from the one we present that would also work well.
>
> Main comparison is with other single parameter methods; but adaptive multi-parameter methods are as easy to implement in practice and should be included.
>
> - While we agree it would be interesting to compare against optimisers with two or more hyperparameters. We do not think this would be a fair comparison and our computation resources would not allow us to do this for a significant number of methods.
>
> Currently while updating AOVs, if the target loss value has not been reached within the K'th section of training, it is updated to a half-way point. This seems quite arbitrary; I agree that the general motivation is correct - that in either case (in Algo 2), you want loss to continuously decrease, but why should l^{k+1} be the half-way point and not l^k + some other \eps ? Can you discuss the reasoning behind this
>
> - Our method for increasing the AOVs (Algorithm.2 9&10) was based on the non-stochastic scheme detailed in Revisiting the Polyak Step Size 2022 Hazan and Kakade. While the previous version of this paper contained sign errors, this appears to now have been corrected on arxiv as of two weeks ago. We have included a citation of the corrected version in our revised paper. Note that, in contrast to our work, this prior work considers the non-stochastic case. Hence, its direct use is computationally prohibitive in deep learning settings. In the convex and non-stochastic case, the AOV is always a lower bound on f(star), hence there is no need to decrease the AOVs. The scheme for decreasing the AOVs (algorithm.2 6&7) simply selects the AOV to be half-way between the past previous AOV that was “reached” and the value that wasn’t. As we avoid fixed \eps our method is invariant to scalings of the loss function. We did try simply backtracking to the last AOV that was not reached, which performed slightly worse. We have added an appendix of unsuccessful ideas to the paper.
>
> How fast is "too fast" for trending towards optimal loss. How do you threshold/measure this?
>
> - Ideally you want the loss to begin to plateau before updating the AOVs. Potentially, we could have used a parameterised criterion to try to update the AOV’s once the loss began to plateau. However it is likely these would need to be tuned per task. We find the simple approach of using 4 AOV updates equally split through training is enough to produce good results on the numerous tasks we investigated.
>
> Worst case bound using ALI-G is actually a good idea in this case. Does ALI-G+ perform better always? Can you comment in the justification section above Section 5
>
> - To the contrary, we expect ALI-G to perform better than ALI-G+ for settings where the interpolation assumption holds. ALI-G+ adjusting AOV away from their true value will be detrimental in this setting.
>
> For Figure 1 - ALI-G+ underperforms compared to SLS and PAL for higher rank problems. Even iterations to reach 10^-4 is better in the case of PAL and SLS. Does K affect this? Can you do more updates initially?
>
> - Yes, it is likely regular ALI-G or K=1 would be best here as the fstar = 0.
>
> Again for Figure 2 - saying it approaches the same val loss seems like a way to explain out of "ALI-G+ fails to achieve the same training loss as the line search methods". Are you only testing for validation performance throughout or training performance as well?
>
> - These plots were generated using the official Painless Stochastic Gradient: Interpolation, Line-Search, and Convergence Rates code which does not track training accuracy. For interpolating problems, ALI-G, that has all AOVs fixed at 0 would be a better choice of Algorithm.
>
> "Adam and Adabound also include momentum like terms which we leave at their default settings." - what are these settings and how may these affect your results?
>
> - Adam β1=0.9, Adabound β1=0.9. For a fair comparison we alter one hyperparameter per optimiser. Hence we would suggest altering these hyperparameters would fall out of the scope of our investigation as we already tune the learning rate for these optimisers.
>
> Earlier the claim was that ALI-G+ is "robust" to choices of K. Seems like reducing K helps from the results as mentioned in the caption of Table 1. What if you reduce it further to (1,5] ? Abstract/intro can be changed to acknowledge this understanding.
>
> We have added results for K=3 in the appendix.

---

### Author Response · Authors · 2022-08-26
**Uploaded revised version of paper**

Thank you for your reviews, We have uploaded a revised version of the paper and anonymised PyTorch code.

---

### Decision · Action_Editors · 2022-09-08

**Recommendation:** Accept as is

**Comment:**

The paper proposes a optimization technique extending ALI-G to non-overparametrized regime, named ALI-G+. This paper is well written and motivated, with results in this restricted setting promising enough to appeal to the audience of TMLR. With reviewers' feedback, the authors have clarified the submission and added new experiments to strengthen it. Therefore, along with the majority of the reviewers, I recommend this paper for acceptance.